# Do Age, Gender and Poor Diet Influence the Higher Prevalence of Nomophobia among Young People?

**DOI:** 10.3390/ijerph17103697

**Published:** 2020-05-24

**Authors:** Antonio-José Moreno-Guerrero, Inmaculada Aznar-Díaz, Pilar Cáceres-Reche, Antonio-Manuel Rodríguez-García

**Affiliations:** Department of Didactics and School Organization, University of Granada, 18071 Granada, Spain; ajmoreno@ugr.es (A.-J.M.-G.); iaznar@ugr.es (I.A.-D.); caceres@ugr.es (P.C.-R.)

**Keywords:** nomophobia, smartphone, addictions, teenagers, youth, eating habits

## Abstract

The use of Information and Communication Technologies (ICT) is generating the emergence of new pathologies such as nomophobia. The aim of this research was to analyze the prevalence of nomophobia among young people, as well as to check whether the level of nomophobia is higher in males or females and in those students who claim to have less healthy nutrition due to the use of their mobile phones. The research method was based on a correlational and predictive design with a quantitative methodology. The measurement tool used is the Nomophobia Questionnaire (NMP-Q). The participating sample was 1743 students between 12 and 20 years old from different educational stages of the Autonomous City of Ceuta (Spain). The results show that highest rates of nomophobia were found in relation to the inability to communicate and contact others immediately. About gender, women have higher rates of nomophobia than men. In relation to age, no significant differences were found; thus, the problem may affect all ages equally. Finally, students who think that their smartphone use is detrimental to their good nutrition show higher levels on the scale provided.

## 1. Introduction

Uncontrolled and irresponsible use of technology is increasingly common among the population [1,2] and, more specifically, among young people [3,4,5]. The smartphone has become the preferred device for accessing the Internet and other communication services, social networks, and supermarkets, among others [6], due mainly to the many possibilities it offers [7,8] and its easy portability [9]. In this way, the smartphone becomes a “buddy” that accompanies us anywhere at any time [10].

Recent research shows that excessive use of these devices can lead to absolute dependence of the user on the object [11,12,13,14] and addictive behavior [15]. This is manifested by the need to use the phone more and more [16,17,18], exhibiting sadness, depression, anger, irritability, restlessness, tension or nervousness when the phone is not available [5]. At the same time, other researchers point out that a new phobia is currently developing caused by this, which manifests itself when people do not immediately have their smartphone or cannot make use of the different possibilities it offers [13,19].

Nomophobia (non-mobile-phobia) is a behavioral disorder that is beginning to develop and be studied in this century, being considered a syndrome of problematic use of digital media [20] originated by four main causes [21,22,23]: (1) not being able to communicate with other people; (2) losing connectedness; (3) not being able to access; and (4) giving up convenience. Some of the most significant characteristics that describe the person with nomophobia are [24,25]:To constantly, daily spend considerable time on one’s mobile phone, and to always carry a charger with oneself.To feel anxious and nervous at the thought of losing one’s own handset; or when the mobile phone is not available nearby, misplaced, or cannot be used because of lack of network coverage, flattened battery, and/or lack of credit.To avoid as much as possible the places and situations in which the use of the device is banned (such as public transit, restaurants, theaters, and airports).To look at the phone’s screen to see whether messages or calls have been received.To keep the mobile phone always switched on (24 h a day).To sleep with the mobile device in bed.To have few social face-to-face interactions with humans which would lead to anxiety and stress and to prefer to communicate using the new technologies.To incur debts or great expense from using the mobile phone.

Although current research is in its early stages [5], people who show higher levels of nomophobia have a greater need to constantly check their mobile device notifications. This, in turn, affects the quality of sleep, eating, health, productivity, concentration, and performance [26,27,28,29,30,31,32].

On the other hand, the development of this phobia and dependence generates greater symptoms of anxiety and depression among those with a higher rate of nomophobia, since they tend to show anxious behavior if their contacts do not respond to them instantly or if they do not talk to them [3]. Some researchers evidence these users also tend to show aggressive behavior, especially when they cannot make immediate use of the device [33] and worse social skills when they interact with other people face to face [34], as well as worse academic performance, lower work productivity or greater deficits in their learning [28,29].

However, since studies are currently few [5], there is no specific profile of the individuals who present it [35]. Up to now, we know that this phobia is characterized by the fear of not being able to communicate, losing connection, loneliness, and loss of comfort [36]. On the other hand, levels of nomophobia have a strong, positive, and significant relationship with the variables of problematic Internet use, anxiety about social appearance, and dependence on social media [37]. Other studies indicate that people with this problem take on maladaptive coping strategies when faced with stress [38,39,40]. On the other hand, words related to memory, self, and proximity seeking are actually more common in the group with higher levels of nomophobia compared to those with lower levels of nomophobia [41]. In addition, other researchers have identified three symptomatic factors that are related to this problem: anxiety, compulsive smartphone use, and feelings of panic [42]. Other researchers have shown that nomophobia can lead to stress through social threat when there is uncertainty or lack of control [43].

This problem seems to affect various populations, especially young people, whether they are university students [38,39,40] or at other educational stages [44,45], especially when they cannot communicate and are disconnected from their virtual identity. In regards to gender, some studies indicate that women show greater symptoms of this phobia due to their higher dependence on their smartphone [39]. Others, however, found no significant differences between men and women [46,47,48,49].

In addition, addictive behavior can lead to health problems and the acquisition of other bad habits in the subject [50] such as not being physically active, poor nutrition [1], chronic tension in the neck and eyes, blurred vision, or problems in the execution of certain movements [51].

Currently, there are few studies investigating the relationship between mobile phone addiction and adolescent diets and eating behaviors [52]. However, existing research concludes that levels of addiction affect students’ eating habits and their own diets [53]. Students with high levels of addiction tend to skip meals [54], which can increase the feeling of being full [55]. Students with low levels of addictions tend to consume more vegetables, milk, and yogurt than those with high levels of addictions. In contrast, those with high levels of addiction consume fewer vegetables, fruits, and dairy products [56,57,58]. A poor diet, caused by mobile phone use, can lead to chronic diseases in adulthood [59,60]. In other words, excessive use of mobile phones can lead to diet problems in students, especially because overuse leads to consumption of fast food, increased body weight, increased body mass index, and decreased physical activity [61,62,63].

In relation to gender, there are no conclusive results on the influence of gender on levels of nomophobia. On the one hand, there are studies that show differences between men and women in relation to levels of nomophobia [64]. In these cases, women have higher levels of nomophobia than men [19], but not in all dimensions. On the other hand, there is no difference in relation to care [65]. In contrast, other studies show that there are no significant differences in nomophobia between gender, and further research is needed [66].

Nomophobia is one of the diseases of the 21st century [67]. Currently, there is not a great scientific production on the subject [5], thus it is necessary to continue research in this reality of the current century. In this sense, this study aimed to analyze the prevalence of nomophobia among adolescents, taking into account age and gender, as well as to verify whether the level of nomophobia is higher in those students who claim to have a worse diet due to overuse or misuse of their mobile phones.

## 2. Materials and Methods

To respond to the general objective, we applied a descriptive, correlational, and transversal methodological design through a quantitative methodology [68]. Therefore, we posed the following research questions:What is the level of nomophobia in the population studied?Do age and gender influence the level of nomophobia?Is nomophobia higher in the population with worse eating habits?

### 2.1. Participants

The study was carried out with students from the Autonomous City of Ceuta, located in the south of Spain, on the African continent and whose size amounts to *N* = 13,721 students from different educational stages: compulsory secondary education, baccalaureate, vocational training, and university studies (University of Granada, Campus of Ceuta). For the selection of the sample, a simple random sampling technique was used with an estimated percentage of 50%, a margin of error of 3%, and a confidence level of 99%.

The final sample was *n* = 1743, with 43.3% male students (755) and 56.7% female students (988). The ages of the participants ranged from 12 to 14 years of age (31.5%), 15 to 17 years of age (45%), 18 to 20 years of age (9.9%), and over 20 years of age (13.6%), from various educational stages, including compulsory secondary education (50.9%), vocational training (6.4%), high school (27%), bachelor’s degree (9.4%), and master’s degree (6.4%).

### 2.2. Measurement

The instrument used for this study is a questionnaire called Nomophobia Questionnaire (NMP-Q), created by Yildirim and Correia [23] and adapted to the Spanish context by other researchers [69,70]. This instrument consists of 27 items, distributed in four dimensions: Dimension I, not being able to communicate (6 items); Dimension II, losing connectedness (5 items); Dimension III, not being able to access information (4 items); and Dimension IV, giving up convenience (5 items), in addition to the socio-educational dimension (7 items).

The items that make up Dimensions I–IV of the NMP-Q questionnaire conform to a Likert-type scale of 7 levels, where 1 is totally in agreement and 7 totally in disagreement. The scores obtained can be between 20 and 140 points. Scores closer to 20 show high levels of nomophobia, while scores closer to 140 denote low levels of nomophobia. The socio-educational dimension is made up of items with a Likert type scale and dichotomous questions (yes/no). No missing values are shown in the results, since all the items of the questionnaire have been established as compulsory. In this dimension, we asked students whether mobile phone use directly affected their diet, specifically their eating habits.

The validity and reliability of the questionnaire is checked against the data presented in [69,70], as the authors themselves conducted various statistical tests. An example of this is Cronbach’s alpha values for each of the dimensions: Dimension I, unable to communicate (*α* = 0.939); Dimension II, losing connectivity (*α* = 0.874); Dimension III, not being able to access information (*α* = 0.827); and Dimension IV, giving up convenience (*α* = 0.819). At a general level, the Cronbach’s alpha result is 0.945.

### 2.3. Study Variables

The study variables were the dependent variables, formed by the different items that make up the questionnaire. The results of these variables were grouped into four dimensions: not being able to communicate (NTCO), losing connectedness (LOCO), not being able to access information (NAIN), and giving up convenience (GUCO). The independent variables established for this study were food (FEED) (Do you think that your use of your mobile phone affects your eating habits?), gender (GEN), and age (AGE).

### 2.4. Procedure

This research began with an exhaustive analysis of the scientific production on the subject to be studied. In this search, the instrument was determined according to the needs of the study. Subsequently, contact was established with the different educational centers where the teaching of compulsory secondary education, baccalaureate, vocational training, and university education in the Autonomous City of Ceuta is carried out. In this contact, collaboration was requested, as well as initiating the necessary protocols to request the required permits, in those centers that asked us to do so, by means of a written document endorsed by the research group.

Once the instrument had been determined and the request for collaboration had begun, the Google form was drawn up with the instrument chosen. The students who participated did so voluntarily and anonymously, with the intention of achieving the highest degree of sincerity in their responses. No students refused to participate in the study, showing themselves to be collaborative throughout the process. The same happened with the volunteers in the data collection, who clarified and solved all the doubts that arose in this phase of the study.

The collection of information was carried out through a single-blind process, so the participants were not shown the objectives and purposes of the research, in order not to generate expectations, reactivity, and social desirability among them. The data collection was carried out in the first week of November, 2019. The completion of the questionnaire took about 10 min, thus avoiding the fatigue of the participants.

### 2.5. Data Analysis

To perform the statistical analysis, we selected the IBM SPSS program, version 25. We first performed several tests to check the assumptions of linearity, independence, normality, homoscedasticity, residue analysis, and non-collinearity. The results obtained show that all the assumptions are strictly adhered to, except those of homoscedasticity and normality of the general linear model in certain variables, given that the *p*-values (sig.) of each of the dimensions are significant because they are less than the critical value *α* = 0.05, i.e., they do not follow the normal distribution at the 95% confidence level, this fact being confirmed by the visual analysis of the histograms, which showed distributions skewed to the right in some cases and to the left in others. Therefore, in these cases, we applied non-parametric statistical tests when appropriate.

We developed several statistical tests to compare the population averages of the scores of each of the dimensions with respect to gender and age group. We then conducted a factor analysis to find out what the response structures of the surveyed sample look like and to check if the dimension structures are equivalent. In the test of average comparison of each dimension according to gender and age, although they do not present an asymptotically normal distribution, the observations being greater than 100, the t-student statistical test was applied. Subsequently, the ANOVA test and the Welch test, where applicable, were performed on the dimensions of the age groups. A proportionality test was followed to find out the students’ nomophobia levels, using the *t*-student statistical test. Finally, the Chi-square test was applied to find out how nomophobia affects the diet.

## 3. Results

In Table 1, the descriptions of the questions answered by the sample of students are located. As the scores for these questions are from one to seven, the minimum and maximum values are one and seven, respectively, for all and the averages are located approximately in the middle zone of the scores from one to seven with a variation (SD) of less than approximately two.

In the test comparing means for each dimension according to gender, the results show a *p*-value less than the critical value at 95% confidence level for the dimensions “Cannot communicate” and “Comfort waiver”, being in this case the significant tests; therefore, “Cannot communicate” and “Comfort waiver” are more painful for female students than for male students with a 95% confidence level. For the other dimensions there is no significant difference (Table 2).

In Table 3, we present the summary of means, SD, confidence interval, variance homogeneity, and ANOVA test. On the one hand, the scores achieved in each established age group do not show large differences among them, if we directly observe the mean obtained. The same is true of the standard deviation. This means that the levels of nomophobia in each of the established age groups show even response and actions.

On the other hand, the ANOVA test was then applied to compare the scores of each dimension according to age groups. With this action, we wanted to confirm the data observed in the table and to corroborate the results obtained previously, as shown in Table 3. This analysis shows that the values obtained in the dimensions “Inability to communicate” and “Giving up comfort” in relation to the different age groups are similar, given that their *p*-values are greater than 0.05. For these cases, the ANOVA test was performed. On the other hand, for the scores of the dimensions “Loss of connection” and “Not being able to access information”, their corresponding *p*-values are lower than the value of 0.05; thus, in these cases, the Welch statistical test was also applied. The results achieved in both statistical tests show that in all cases the values are not significant at the 95% confidence level, because their corresponding *p*-values are lower than 0.05. Therefore, it is determined that there is no significant difference between the age group averages in each of the scores, confirming the above.

In this phase of making a contrast between them, they are in complete agreement with what each of the dimensions says or they are in disagreement. The students answered a questionnaire with a set of items and each of them has the seven-point Likert answer structure. If we classify the scores of each of the dimensions into two categories, those who totally agree and those who totally disagree, for example, for the dimension “Unable to communicate”, we classify the scores less than or equal to the value 23 as those who totally agree and the other category, those who totally disagree, as values greater than or equal to 24 (including those who are indifferent) (i.e., values less than 24 as those who are indifferent and values greater than 24 as those who totally disagree), in proportions according to Table 4, starting from left to right, we have 59.2% of accumulated percentage up to value 23, those who totally agree with being afraid of being without a telephone (nomophobia), that is “not being able to communicate” with their relatives and friends and their complement is 100% − 59.2% = 40.8% those who totally disagree with this dimension. In this situation, the corresponding percentage where we must consider the category is considered totally in agreement are highlighted in yellow for each dimension, and obviously the complementary corresponds to the other category of totally in disagreement.

These proportions or percentages of each of the categories for each of the dimensions are summarized in Table 5. This table shows the percentage of the sample studied that totally agrees and the percentage that totally disagrees for each of the dimensions. For the first dimension, the majority agrees, but for the last three dimensions the highest percentage corresponds to disagreeing completely.

In the proportionality contrast test, we tried to find out, in general terms, if the young students in Ceuta have a certain level of nomophobia, i.e., if they are afraid of losing their mobile phone and “Not being able to communicate”; fear of losing their mobile phone and “Loss of connection” with the media, social networks and e-mail; fear of losing their mobile phone and “Not being able to access information” accessing information through their smartphone; and fear of losing their mobile phone and “Giving up comfort” due to problems with their battery, balance. The results show that only in the first dimension, “Not being able to communicate”, there is nomophobia in students over 12 years old in Ceuta (Table 6).

Table 7 presents the data collected considering those who have answered yes and no in the variable of eating habits (Do you think that your use of your mobile phone affects your eating habits?). As we can see, 213 participants answered yes, distributed in 45.1% of men and 54.9% of women. In addition, of the 213 who answered yes, 39.4% were between 12 and 14 years old, 39.9% were between 15 and 17 years old, 7% were between 18 and 20 years old, and 13.6% were over 20 years old. In the contrast on whether the level of nomophobia affects the eating habits, it is observed that the proportion or percentage of agreeing totally is always greater when disagreeing totally for all dimensions; thus, we can say that nomophobia affects the eating habits significantly (Table 7).

## 4. Discussion

The use of information and communication technologies is advancing incessantly in all corners of society, interfering almost totally in all the actions we develop, 24 h a day, generating new social routines, focused mainly in the social, personal, and labor fields [1,2,3,4,5,6,7,8,9,10].

All this promotes benefits in our daily tasks, but it is also beginning to generate new pathologies, among which are phobias and addictions. An example of this is the emergence of the concept of nomophobia, which refers to the anxiety caused in people by not being able to access the smartphone at a certain time [11,12,13,14,15,16,17,18,19,20,21,22,23].

The present research analyzed the effect nomophobia has according to the gender and age of the person, the level of nomophobia of the students, besides knowing how this pathology influences the diet of the students in the Autonomous City of Ceuta.

Analyzing each of the study dimensions in depth, on the one hand, the results show that the levels of nomophobia among the population studied are in an intermediate position. However, the impossibility of contacting or being contacted by family and friends is the variable with the highest prevalence. This fact reaffirms the findings found by other researchers [38,40,44,45]. On the other hand, levels of nomophobia are lower in relation to other dimensions, as shown by other researchers [3,21,22,23].

In relation to gender, there is a direct relationship between not being able to communicate or giving up comfort, with nomophobia affecting women more directly than men. This does not occur with the loss of connection and not being able to access information, where no significant gender differences are observed. These data are in line with other research [19,40,63], which confirms that the levels of nomophobia are higher in women than in men. However, these results are contradicted by other research that found no significant differences between the two genders [49,65,66].

Age is not an influential element, since levels of nomophobia are equal in all established age groups. Therefore, no significant differences are shown. Even so, if we look only at the averages achieved, people over 20 years of age present lower levels of nomophobia than students between the ages of 15 and 20. Even so, the levels presented at all established ages are medium-low.

Finally, it can be stated that the levels of nomophobia in most of the dimensions analyzed, i.e., the inability to communicate, loss of connection, inability to access information, and relinquishment of comfort, have a significant effect on eating, which means that students with higher levels of nomophobia in the sample studied have their eating habits affected. This would serve to include new research on the areas of nutrition in relation to mobile phone use, in addition to confirming what other researchers have already stated [50,51,52,53,54,55,56,57,58,59,60,61,62,63]: high levels of mobile phone addiction lead to changes in students’ eating habits, causing changes in the type of food they eat, in addition to generating possible future health problems.

## 5. Conclusions

Once the findings of the study are presented, we proceed to delimit the conclusions we reached. Firstly, the female participants have higher levels on the scale provided compared to men. In terms of age, no significant differences were found, so that this problem may affect both adolescents and young adults equally.

The inability to contact and communicate immediately with other people is the main cause that generates greater prevalence of this phobia in the sample studied. In addition, those people in the sample who, from their own point of view, claim to have a worse diet due to the excessive use they make of their smartphone show higher levels of nomophobia.

The interest of this study lies in offering a vision to the educational administration and society in general about the prevalence of this problem among adolescents and young people who are in school at different educational stages. In addition, the study shows the need for the educational institution to develop prevention and habit improvement programs both in the use of technology and in the eating habits of students.

The prospective study aims to offer the various administrations, especially the education administration, a state of affairs regarding nomophobia among schoolchildren over 12 years of age, especially in terms of eating habits, so that they can establish programs to raise awareness and educate people about healthy eating habits in relation to the use of smartphones.

Finally, we must point out that this research presents different limitations that will be taken into consideration for future publications. First, we must point out those inherent to the instrument supplied to the sample, given that some questions were asked about the person’s subjective perception (Yes/No) with respect to various variables, one of which was food. On the other hand, the bureaucratic process to be able to access such a large sample was complicated. In some centers, asking for verbal permission was more than enough. In others, we had to make written presentations and record them. Another limitation we faced in this research was the equipment needed to be able to answer the questionnaire. In some centers, the technological resources were not enough. We solved this by providing the researchers with their own material or by inviting them to use the devices that the students brought with them. As future lines of research, we aim to develop this study in other contexts and provide other standardized questionnaires, in order to contrast and confirm the results obtained in this study and to expand knowledge in this growing field.

## Figures and Tables

**Table 1 ijerph-17-03697-t001:** Descriptive analysis.

Item	*N*	Min	Max	*X*	SD
I would worry about not being able to communicate instantly with my family and/or friends.	1743	1	7	3.22	1.987
I’d be worried that my family and/or friends wouldn’t be able to contact me.	1743	1	7	3.09	1.991
I would be nervous about not being able to receive text messages or calls.	1743	1	7	4.20	2.124
I would be uneasy about not being able to keep in touch with my family and/or friends.	1743	1	7	3.33	2.008
I would be nervous about not being able to know if someone has tried to contact me.	1743	1	7	3.92	2.026
I would worry that I have stopped being in constant contact with my family and/or friends.	1743	1	7	3.69	1.975
I would be nervous about being disconnected from my virtual identity.	1743	1	7	4.68	2.053
I would feel bad for not being able to keep up with what’s going on in the media and social networks.	1743	1	7	4.42	2.102
I would feel uncomfortable not being able to consult notifications about my connections and virtual networks.	1743	1	7	4.46	2.040
I would be overwhelmed by not being able to check for new emails.	1743	1	7	4.92	2.107
I’d feel weird because I wouldn’t know what to do.	1743	1	7	4.38	2.115
I’d feel bad if I couldn’t access the information on my smartphone at any time.	1743	1	7	4.06	1.979
I’d be upset if I couldn’t get information through my smartphone when I wanted to.	1743	1	7	3.68	1.975
I would be nervous if I could not access the news (e.g., events, weather forecast, etc.) through my smartphone.	1743	1	7	4.47	2.043
I’d be upset if I couldn’t use my smartphone and/or its applications when I wanted.	1743	1	7	3.78	2.026
I’d be scared if my smartphone ran out of battery.	1743	1	7	4.77	2.134
I’d get something if I was about to run out of credit or reach my monthly spending limit.	1743	1	7	4.92	2.161
If I run out of data or can’t connect to a Wi-Fi network, I’m constantly checking to see if I’ve recovered the signal or can find a network.	1743	1	7	3.94	2.160
If I couldn’t use my smartphone, I’d be afraid I’d get stuck somewhere.	1743	1	7	3.79	2.170
If I couldn’t check my smartphone for a while, I’d feel like doing it.	1743	1	7	4.05	2.077

Source: Own elaboration.

**Table 2 ijerph-17-03697-t002:** Statistical testing by gender.

Dimension	Gender	Mean	SD	Normality Test *p*-Value	Equal Variance Test *p*-Value	Average Equality Test *p*-Value	95% Confidence Interval of the Difference	
							Lower	Upper
Not being able to communicate	Man	22.00	9.384	0.000	0.932	0.034	0.070	1.845
	Woman	21.04	9.336	0.000				
Losing connectedness	Man	22.59	8.298	0.000	0.520	0.222	−1.280	0.297
	Woman	23.08	8.334	0.000				
Not being able to access information	Man	15.96	6.587	0.000	0.759	0.873	−0.676	0.574
	Woman	16.02	6.598	0.000				
Giving up convenience	Man	21.90	7.954	0.000	0.232	0.049	0.002	1.521
	Woman	21.13	8.054	0.000				

Source: Own elaboration.

**Table 3 ijerph-17-03697-t003:** Summary of means, SD, confidence interval, variance homogeneity, and ANOVA test of the dimension according to age.

Dimension	Age	Total	Mean	SD	95% Confidence Interval of the Difference		Equal Variance *p*-Value	Mean difference *p*-Value	Test
					Lower	Upper			
Not being able to communicate	12–14	549	21.65	9.828	20.82	22.47	0.111	0.088	ANOVA
	15–17	785	21.13	9.073	20.49	21.76			
	18–20	172	20.64	9.172	19.26	22.02			
	+20	237	22.68	9.295	21.49	23.87			
Losing connectedness	12–14	549	22.57	8.900	21.82	23.32			
	15–17	785	22.89	7.787	22.34	23.43	0.000	0.450	Welch
	18–20	172	22.63	8.374	21.37	23.89			
	+20	237	23.64	8.594	22.54	24.74			
Not being able to access information	12–14	549	16.02	6.963	15.43	16.60			
	15–17	785	15.71	6.281	15.27	16.15	0.001	0.053	Welch
	18–20	172	15.73	6.222	14.80	16.67			
	+ 20	237	17.08	6.892	16.19	17.96			
Giving up convenience	12–14	549	21.56	8.350	20.86	22.26			
	15–17	785	21.05	7.764	20.51	21.60	0.068	0.062	ANOVA
	18–20	172	21.39	8.024	20.18	22.60			
	+ 20	237	22.65	7.979	21.62	23.67			

Source: Own elaboration.

**Table 4 ijerph-17-03697-t004:** Proportion of dimensional scores.

Not Being Able to Communicate				Losing Connectedness				Not Being Able to Access Information				Giving up Convenience			
V	Fr	%	% A	V	Fr	%	% A	V	Fr	%	% A	V	Fr	%	% A
6	85	4.9	4.9	5	45	2.6	2.6	4	75	4.3	4.3	5	53	3.0	3.0
7	17	1.0	5.9	6	16	0.9	3.5	5	29	1.7	6.0	6	13	0.7	3.8
8	43	2.5	8.3	7	18	1.0	4.5	6	34	2.0	7.9	7	29	1.7	5.5
9	42	2.4	10.7	8	25	1.4	6.0	7	64	3.7	11.6	8	25	1.4	6.9
10	46	2.6	13.4	9	25	1.4	7.4	8	63	3.6	15.2	9	32	1.8	8.7
11	37	2.1	15.5	10	29	1.7	9.1	9	45	2.6	17.8	10	39	2.2	11.0
12	78	4.5	20.0	11	50	2.9	11.9	10	79	4.5	22.3	11	37	2.1	13.1
13	57	3.3	23.0	12	37	2.1	14.1	11	75	4.3	26.6	12	38	2.2	15.3
14	76	4.4	27.6	13	35	2.0	16.1	12	99	5.7	32.3	13	50	2.9	18.1
15	57	3.3	30.9	14	39	2.2	18.3	13	81	4.6	36.9	14	49	2.8	20.9
16	49	2.8	33.7	15	53	3.0	21.3	14	104	6.0	42.9	15	64	3.7	24.6
17	53	3.0	36.7	16	38	2.2	23.5	15	96	5.5	48.4	16	65	3.7	28.3
18	82	4.7	41.4	17	72	4.1	27.7	16	129	7.4	55.8	17	73	4.2	32.5
19	60	3.4	44.9	18	51	2.9	30.6	17	90	5.2	61.0	18	59	3.4	35.9
20	69	4.0	48.8	19	50	2.9	33.4	18	67	3.8	64.8	19	50	2.9	38.8
21	66	3.8	52.6	20	89	5.1	38.6	19	68	3.9	68.7	20	96	5.5	44.3
22	54	3.1	55.7	21	58	3.3	41.9	20	71	4.1	72.8	21	81	4.6	48.9
23	61	3.5	59.2	22	71	4.1	46.0	21	54	3.1	75.9	22	77	4.4	53.4
24	89	5.1	64.3	23	77	4.4	50.4	22	80	4.6	80.5	23	87	5.0	58.3
25	59	3.4	67.7	24	52	3.0	53.4	23	58	3.3	83.8	24	57	3.3	61.6
26	48	2.8	70.5	25	73	4.2	57.5	24	61	3.5	87.3	25	87	5.0	66.6
27	44	2.5	73.0	26	57	3.3	60.8	25	55	3.2	90.5	26	62	3.6	70.2
28	45	2.6	75.6	27	85	4.9	65.7	26	36	2.1	92.5	27	53	3.0	73.2
29	43	2.5	78.0	28	63	3.6	69.3	27	37	2.1	94.7	28	63	3.6	76.8
30	57	3.3	81.3	29	76	4.4	73.7	28	93	5.3	100.0	29	72	4.1	81.0
31	38	2.2	83.5	30	64	3.7	77.3					30	68	3.9	84.9
32	38	2.2	85.7	31	69	4.0	81.3					31	57	3.3	88.1
33	27	1.5	87.2	32	77	4.4	85.7					32	46	2.6	90.8
34	25	1.4	88.6	33	66	3.8	89.5					33	47	2.7	93.5
35	36	2.1	90.7	34	55	3.2	92.7					34	39	2.2	95.7
36	41	2.4	93.1	35	128	7.3	100.0					35	75	4.3	100.0
37	23	1.3	94.4												
38	21	1.2	95.6												
39	18	1.0	96.6												
40	17	1.0	97.6												
41	9	0.5	98.1												
42	33	1.9	100.0												

Note: Value (V); Frequency (Fr); Percentage (%); Cumulative percentage (% A).

**Table 5 ijerph-17-03697-t005:** Percentage agreeing or disagreeing with the corresponding dimension.

Dimension	Not being able to Communicate		Losing Connectedness		Not Being Able to Access Information		Giving up Convenience	
	Fr	% V	Fr	% V	Fr	% V	Fr	% V
Strongly agree	772	44.3	1160	66.6	899	51.6	1067	61.2
Strongly disagree	971	59.2	583	33.4	844	48.4	676	38.8
Total	1743	100.0	1743	100.0	1743	100.0	1743	100.0

Note: Frequency (Fr); Valid percentage (% V).

**Table 6 ijerph-17-03697-t006:** Proportionality test.

Test Value = 0.51	
Dimension	*p*-Value (Bilateral)
Not being able to communicate	0.000
Losing connectedness	0.994
Not being able to access information	0.984
Giving up convenience	0.990

Source: Own elaboration.

**Table 7 ijerph-17-03697-t007:** Cross table and Chi-square test.

Eating Habits	Dimension		Strongly Agree	Strongly Disagree	Count	*p*-Value
Yes	Not being able to communicate	Fr	65	148	213	0.000
		%	9.1%	14.3%	12.2%	
Not		Fr	646	884	1530	
		%	90.9%	85.7%	87.8%	
Yes	Losing connectedness	Fr	107	106	213	0.000
		%	9.2%	18.2%	12.2%	
Not		Fr	1053	477	1530	
		%	90.8%	81.8%	87.8%	
Yes	Not being able to access information	Fr	83	130	213	0.000
		%	9.2%	15.4%	12.2%	
Not		Fr	816	714	1530	
		%	90.8%	84.6%	87.8%	
Yes	Giving up convenience	Fr	98	115	213	0.000
		%	9.2%	17.0%	12.2%	
Not		Fr	969	561	1530	
		%	90.8%	83.0%	87.8%	

Note: Frequency (Fr); Percentage (%).

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
