# Peer review of "Do Age, Gender and Poor Diet Influence the Higher Prevalence of Nomophobia among Young People?"

_ijerph, 2020, doi:10.3390/ijerph17103697_

Round 1

Reviewer 1 Report

The paper is relevant and I'm very interesting in the topic.

However, this study raises serious doubts. I could understand, for example, if the authors would like to study only the incidence of nomophobia in a given population.

However, in order to question whether this obsession is superior in young people with eating problems, it is necessary to know if they have problems before the compulsive use of devices. That is, the use of these could be a way to reduce anxiety.

Perhaps it would be useful to use a questionnaire in parallel to assess food habits retrospectively. There are some that can be used with this objective.

Facing these objectives, my first suggestion is rejecting the paper.

Anyway, if you are able to study only the incidence of nomophobia in that population, I think you can present an interesting study, namely to understand the activities of daily live, including food habits, in the participants. Of course this means a major revision.

Author Response

The paper is relevant and I'm very interesting in the topic.However, this study raises serious doubts. I could understand, for example, if the authors would like to study only the incidence of nomophobia in a given population. 

Response: Dear Reviewer, thank you for your ratings and comments. We hope to improve the quality of the manuscript with the proposed aspects. 

However, in order to question whether this obsession is superior in young people with eating problems, it is necessary to know if they have problems before the compulsive use of devices. That is, the use of these could be a way to reduce anxiety. 

Response: Dear reviewer, in this study we tried to analyze several factors in relation to nomophobia. In relation to diet, it is true that we did not pass a specific questionnaire, we only asked the question of whether diet affected their eating habits, that is, whether it affected changes in routines, changes in types of food,... Although this is one of the aspects included within the study, there are other elements, with age and gender, that have been analysed and studied. The item of healthy nutrition was complementary to the study. 

Perhaps it would be useful to use a questionnaire in parallel to assess food habits retrospectively. There are some that can be used with this objective. 

Response: Dear Reviewer, on this, after receiving your assessments, we agree with you. The fact is that it is now unfeasible to carry it out. Rest assured that in order to expand this study, we will apply complementary questionnaires in the future. 

Facing these objectives, my first suggestion is rejecting the paper. 

Response: Dear reviewer, we have tried to improve the manuscript, expanding the theoretical framework and the discussion mainly, among other aspects. With this we try to adapt the possible deficiencies that the manuscript may have. We consider that very interesting results are obtained, especially in relation to gender and age, in a very specific population context, making use of sampling techniques. 

Anyway, if you are able to study only the incidence of nomophobia in that population, I think you can present an interesting study, namely to understand the activities of daily live, including food habits, in the participants. Of course this means a major revision. 

Response: Dear Reviewer, we hope that we have been able to respond to what you have raised with us. We are sorry if at any time we cannot do so. But rest assured that for future studies, we will bear your words in mind. 

Reviewer 2 Report

Abstract should be written more clearly and soundly. The main idea and findings should be presented. „Autonomous City of Ceuta“ was repeated twice, although the description of the sample was not presented. It is not correct to describe the sample as adolescents, as they were also young adults. It is difficult to understand, what does the „problematic couples“ mean?

English should be improved / edited in Abstract and Introduction. E.g., lines 31-32, 43-44, 45-46. Sentence in lines 62-63 needs references. It is not clear, what does „attachment to anxiety“ (line 72) mean?

Methods are described rather adequately. However, there is very few information about validation of four factors model of the NMP-Q. If this information is correct: „In the second analysis, by varimax rotation, dimension I presented a variance of 22.38%, dimension II a variance of 16.82%, dimension III a variance of 11.87% and dimension IV a variance of 11.59%.” then it means that each factor explains quite small proportion of data variance? And are these results based on the data of this study? If no, then the reference is needed.

Next, what was the exact question for measuring eating habits? In overall, using only one yes/no question for eating habits is one of the most serious limitations of this study. Even more cautious is to make the conclusions such like “their diet is affected by all the dimensions of nomophobia, causing changes in their eating habits”.

Only 213 respondents out of more than 1700 responded ‘yes’ to this question on eating habits (which is not very clear). What was the proportions of female / male who responded with ‘yes’? Does the mean age differ in the two groups (yes / no)?

What is a rationale for presenting the detailed information for every item of NMP-Q? What do results presented in Table 1 and Table 5 add to discussion? Or how do they are important or significant?

Further, authors state that (lines 222-223): “This table shows the percentage of the sample studied that totally agrees (yellow colour) and the percentage that totally disagrees (blue colour) for each of the dimensions.”, but I cannot see colors and do not understand what are they for.

In Table 2 and Table 3 one column is named “Media” – is it correct? Or should it be “Mean”? Results in Table 3 and Table 4 could be combined, e.g. presented in one table.

Discussion is very scarce, presentation of the results of this study in the light of other studies is very limited. As for example, it looks like there are more studies conducted to measure the nomophobia in students and youths, also using the same instruments (NMP-Q)? Why do not make them comparable? Could authors point any findings, knowledge, etc. they add to scientific discussion in a field?

As mentioned earlier, it is very cautious and incorrect to conclude about "effect", as this study is correlational. E.g., "The results show that the female gender is more affected than the male gender in nomophobia values" or, even more, “Food is affected by all dimensions of nomophobia, causing changes in their eating habits.”

Authors did find only one limitation of their study. More sound discussion on methods and their validity is needed. Also, the discussion on future directions is highly desirable.

Author Response

Abstract should be written more clearly and soundly. The main idea and findings should be presented. „Autonomous City of Ceuta“ was repeated twice, although the description of the sample was not presented. It is not correct to describe the sample as adolescents, as they were also young adults. It is difficult to understand, what does the „problematic couples“ mean? 

Response: Dear Reviewer, thank you for your suggestions. We have modified the abstract according to your guidelines. 

English should be improved / edited in Abstract and Introduction. E.g., lines 31-32, 43-44, 45-46. Sentence in lines 62-63 needs references. It is not clear, what does „attachment to anxiety“ (line 72) mean? 

Response: Dear Reviewer, thank you for your suggestions. We have improved the language throughout the paper, especially in the lines you mention. In addition, we have included references to other research. 

Methods are described rather adequately. However, there is very few information about validation of four factors model of the NMP-Q. If this information is correct: „In the second analysis, by varimax rotation, dimension I presented a variance of 22.38%, dimension II a variance of 16.82%, dimension III a variance of 11.87% and dimension IV a variance of 11.59%.” then it means that each factor explains quite small proportion of data variance? And are these results based on the data of this study? If no, then the reference is needed. 

Response: Dear reviewer, we have modified the section referring to the validity and reliability of the instrument. In this case, the instrument has already been validated and made reliable by the authors indicated in the text. We have collected the most relevant values of the study. 

Next, what was the exact question for measuring eating habits? In overall, using only one yes/no question for eating habits is one of the most serious limitations of this study. Even more cautious is to make the conclusions such like “their diet is affected by all the dimensions of nomophobia, causing changes in their eating habits”. 

Response: Dear reviewer, since we can no longer include a questionnaire of our own to assess students' eating habits, we have made modifications to the sentence you indicated, being more cautious in the interpretation of the data. 

Only 213 respondents out of more than 1700 responded ‘yes’ to this question on eating habits (which is not very clear). What was the proportions of female / male who responded with ‘yes’? Does the mean age differ in the two groups (yes / no)? 

Response: Dear reviewer, we have included, in the paragraph above the indicated table, the percentage of both men and women. In addition, the percentage by age. We have also clarified aspects related to the data reached in table 7. 

What is a rationale for presenting the detailed information for every item of NMP-Q? What do results presented in Table 1 and Table 5 add to discussion? Or how do they are important or significant? 

Response: Dear Reviewer, we have included a paragraph in the discussion regarding the results achieved in Table 1. With respect to table 5, what has been done to a response grouping, with the intention of being able to carry out the study of table 7. 

Further, authors state that (lines 222-223): “This table shows the percentage of the sample studied that totally agrees (yellow colour) and the percentage that totally disagrees (blue colour) for each of the dimensions.”, but I cannot see colors and do not understand what are they for. 

Response: Dear Reviewer, thank you for your suggestions. We have modified these issues that you mention. 

In Table 2 and Table 3 one column is named “Media” – is it correct? Or should it be “Mean”? Results in Table 3 and Table 4 could be combined, e.g. presented in one table. 

Response: Dear Reviewer, thank you for your suggestions. We have modified these issues that you mention and have merged tables 3 and 4. 

Discussion is very scarce, presentation of the results of this study in the light of other studies is very limited. As for example, it looks like there are more studies conducted to measure the nomophobia in students and youths, also using the same instruments (NMP-Q)? Why do not make them comparable? Could authors point any findings, knowledge, etc. they add to scientific discussion in a field? 

Response: Dear reviewer, we have carried out an extension of the discussion, referring to the aspects you have indicated. 

As mentioned earlier, it is very cautious and incorrect to conclude about "effect", as this study is correlational. E.g., "The results show that the female gender is more affected than the male gender in nomophobia values" or, even more, “Food is affected by all dimensions of nomophobia, causing changes in their eating habits.” 

Response: Dear Reviewer, thank you for your suggestions. We have modified these issues that you mention by taking more care in the use of language. 

Authors did find only one limitation of their study. More sound discussion on methods and their validity is needed. Also, the discussion on future directions is highly desirable. 

Response: Dear reviewer, we have specified the limitations of the study even more. In addition, we have presented it in an adequate way, given that there are actually 3 limitations in the present research 

Reviewer 3 Report

I consider that the study is interesting and brings novelties to the field of research. Despite this, it is somewhat local, so it would be interesting to compare the results obtained with other similar ones that exist.

Author Response

I consider that the study is interesting and brings novelties to the field of research. Despite this, it is somewhat local, so it would be interesting to compare the results obtained with other similar ones that exist. 

Response: Dear Reviewer, we have taken into account the above. To this end, we have considerably broadened the theoretical framework, and expanded the discussion of the study. 

Round 2

Reviewer 1 Report

The article is much better - the authors introduced many of the suggestions made by the reviewers. I leave two or three additional suggestions in the text that may serve to improve the quality of the paper.

Author Response

The article is much better - the authors introduced many of the suggestions made by the reviewers. I leave two or three additional suggestions in the text that may serve to improve the quality of the paper.

Reply:

Dear reviewer,
We are very grateful for your guidance in improving this work. We have taken note of all the guidelines you have included in the PDF file and have modified them accordingly.

Reviewer 2 Report

I appreciate the authors efforts to adhere to all the comments and suggestions. This helped to make a manuscript more sound and valuable.

There are only several minor issues left:

1) several mistakes or inaccuracies (e.g., no space after dot, different font size in text, etc.);

2) there is no information provided in Authors' reply or in the revised manuscript for this previous comment: "what was the exact question for measuring eating habits?"; Authors say that they do not longer use a questionnaire for this? But they do provide the analysis in Results section and in Discussion on whether nomophobia is related to eating habits... Thus, it means, that they do used a single "yes/no" question for this? Again, what was the exact wording of the question used (for FOOD variable)?

3) Table 4, what is indicated in red color? This should be explained in Notes right after the table.

4) In Discussion, line 305, should it be "eating habits in students" instead of "feeding in students"?

Please, keep on consistent and clear description and interpretation: whether you asked (and provided results) on food/feeding or eating habits?

Author Response

I appreciate the authors efforts to adhere to all the comments and suggestions. This helped to make a manuscript more sound and valuable.

There are only several minor issues left:
Dear reviewer,

Thank you for your comments to improve the work.
1) several mistakes or inaccuracies (e.g., no space after dot, different font size in text, etc.);

We have fixed these mistakes, thank you.

2) there is no information provided in Authors' reply or in the revised manuscript for this previous comment: "what was the exact question for measuring eating habits?"; Authors say that they do not longer use a questionnaire for this? But they do provide the analysis in Results section and in Discussion on whether nomophobia is related to eating habits... Thus, it means, that they do used a single "yes/no" question for this? Again, what was the exact wording of the question used (for FOOD variable)?

We have included the question concerning eating habits.

3) Table 4, what is indicated in red color? This should be explained in Notes right after the table.

We have removed red colour from the table.

4) In Discussion, line 305, should it be "eating habits in students" instead of "feeding in students"?

We've changed "feeding in students" to "eating habits in students"

Please, keep on consistent and clear description and interpretation: whether you asked (and provided results) on food/feeding or eating habits?
